# Evolution of Quorum Sensing in *Pseudomonas aeruginosa* Can Occur via Loss of Function and Regulon Modulation

Priyanikha Jayakumar,[a] Alexandre R. T. Figueiredo,[a,b,c] Rolf Kümmerli[a]

aDepartment of Quantitative Biomedicine, University of Zurich, Zurich, Switzerland
bDepartment of Evolutionary Biology and Environmental Studies, University of Zurich, Zurich, Switzerland
cDepartment of Zoology, University of Oxford, Oxford, United Kingdom

**ABSTRACT** *Pseudomonas aeruginosa* populations evolving in cystic fibrosis lungs, animal hosts, natural environments and in vitro undergo extensive genetic adaption and diversification. A common mutational target is the quorum sensing (QS) system, a three-unit regulatory system that controls the expression of virulence factors and secreted public goods. Three evolutionary scenarios have been advocated to explain selection for QS mutants: (i) disuse of the regulon, (ii) cheating through the exploitation of public goods, or (ii) modulation of the QS regulon. Here, we examine these scenarios by studying a set of 61 QS mutants from an experimental evolution study. We observed nonsynonymous mutations in all three QS systems: Las, Rhl, and *Pseudomonas* Quinolone Signal (PQS). The majority of the Las mutants had large deletions of the Las regulon, resulting in loss of QS function and the inability to produce QS-regulated traits, thus supporting the first or second scenarios. Conversely, phenotypic and gene expression analyses of Rhl mutants support network modulation (third scenario), as these mutants overexpressed the Las and Rhl receptors and showed an altered QS-regulated trait production profile. PQS mutants also showed patterns of regulon modulation leading to strain diversification and phenotypic tradeoffs, where the upregulation of certain QS traits is associated with the downregulation of others. Overall, our results indicate that mutations in the different QS systems lead to diverging effects on the QS trait profile in *P. aeruginosa* populations. These mutations might not only affect the plasticity and diversity of evolved populations but could also impact bacterial fitness and virulence in infections.

**IMPORTANCE** *Pseudomonas aeruginosa* uses quorum sensing (QS), a three-unit multilayered network, to coordinate expression of traits required for growth and virulence in the context of infections. Despite its importance for bacterial fitness, the QS regulon appears to be a common mutational target during long-term adaptation of *P. aeruginosa* in the host, natural environments, and experimental evolutions. This raises questions of why such an important regulatory system is under selection and how mutations change the profile of QS-regulated traits. Here, we examine a set of 61 experimentally evolved QS mutants to address these questions. We found that mutations involving the master regulator, LasR, resulted in an almost complete breakdown of QS, while mutations in RhlR and PqsR resulted in modulations of the regulon, where both the regulon structure and the QS-regulated trait profile changed. Our work reveals that natural selection drives diversification in QS activity patterns in evolving populations.

**KEYWORDS** *Pseudomonas aeruginosa*, evolution, mutation, opportunistic pathogen, quorum sensing, virulence factors

Address correspondence to Priyanikha Jayakumar, priyanikha.jayakumar@uzh.ch, or Rolf Kümmerli, rolf.kuemmerli@uzh.ch.

The authors declare no conflict of interest.

P*seudomonas aeruginosa* is an opportunistic bacterial pathogen responsible for chronic infections, especially in individuals with the genetic disorder cystic fibrosis (CF) (1, 2). *P. aeruginosa* lineages isolated from patients are often characterized by a series of specific mutations, which have been traditionally interpreted as adaptations to the CF lung environment

(3–5). The quorum-sensing regulon is one of the most commonly observed mutational hot spots (4, 6–10). Because *P. aeruginosa* uses QS to regulate the production of a suite of virulence factors and pathogenicity (11–14), it is unclear why mutations in QS regulators (often interpreted as loss-of-function mutations) are selectively favored in an infectious context (15, 16). One possible explanation is that loss of QS function could be beneficial to temper immune responses in chronic infections (4, 17, 18).

QS signaling in *P. aeruginosa* is mediated by two *N*-acyl homoserine lactone (AHL)-dependent systems, the Las and Rhl systems, as well as a third, the *Pseudomonas* Quinolone Signal (PQS) system (14, 19, 20). The signal synthase of each system produces its respective signaling molecule (LasI: 3-oxo-$C_{12}$-HSL; RhlI: $C_4$-HSL, PqsABCD: 2-heptyl-3-hydroxy-4-quinolone) that is recognized by its cognate receptor (LasR, RhlR, and PqsR). Signal-receptor complexes form transcriptional regulators that control the expression of a suite of secreted exoproducts, such as proteases, biosurfactants, toxins, and biofilm structural components. All of these exoproducts are considered virulence factors, harming the host during infections, but they can also have beneficial effects for bacteria in a noninfectious context. For example, proteases can digest extracellular proteins, while certain toxins (e.g., pyocyanin) can catalyze redox processes (21–24). The induction of these exoproducts depends on surpassing a signal threshold concentration, which is often reached at high bacterial population densities. The QS systems are arranged in a hierarchical signaling cascade in which the Las system positively regulates both the Rhl and PQS systems through the binding of Las signal-receptor dimer complex to the promoter regions of RhlR and PqsR, respectively. Additionally, the PQS system can positively regulate the Rhl system, through the binding of PqsE as an alternative ligand to RhlR (25–29), while Rhl in turn inhibits the PQS system (30, 31).

*P. aeruginosa* isolates from chronically infected CF patient lungs frequently contain mutations in the master transcriptional regulator, LasR, thereby influencing the activity of all three QS systems (6, 32, 33). Increased strain diversity in populations harboring mutations in LasR are generally associated with an overall pattern of reduced virulence (16). While initially interpreted as a specific adaptation to the CF lung environment, it has become clear that *lasR* mutants are also selected for in a large range of infection types (34), including chronic wounds (35), corneal infections (36, 37), ventilator-associated pneumonia (30), and infections of the nematode *Caenorhabditis elegans* (18, 38). Additionally, *lasR* mutants are ubiquitous in environments such as hospital sink drains and wet markets (39), and spontaneously evolve under laboratory conditions on agar plates and in liquid, static, and well-mixed environments (33, 38, 40, 41).

Why, then, are *lasR* mutants consistently favored across these different environments? Three competing hypotheses have been advocated. First, mutations in *lasR* lead to a loss of function in QS-regulated phenotypes and are favored because QS is no longer needed in the respective environments, especially during growth in rich medium (42). Second, QS is still required but *lasR* mutants are cheaters that no longer respond to the QS signal. They refrain from producing QS-regulated traits, yet still benefit from the shared pool of secreted QS-regulated traits in the environment (proteases, biosurfactants, toxins) produced by QS wild type cells (16, 30, 31). Third, mutations in *lasR* may modulate the QS regulon itself by either changing its sensitivity or remodeling the hierarchal network as an adaptation to the prevailing conditions (32, 33). The latter hypothesis has been fueled by recent findings that evolved *lasR* mutants have diverse phenotypes and are not necessarily null mutants (6, 18, 43). While these considerations show that most studies are focused on mutations in *lasR*, much less is known about the phenotypes associated with mutations in the Rhl and PQS systems and how they link to the three hypotheses outlined above.

To obtain a deeper understanding of how mutations in all three QS systems affect downstream phenotypes and QS network topology, we used a set of 61 experimentally evolved QS mutants to investigate whether these mutants have lost the ability to produce QS-regulated traits (supporting either the first or the second hypotheses) or show an altered QS-regulated trait expression profile (supporting the third hypothesis). The mutant collection stems from an experimental evolution study performed in our laboratory that focused on the evolution of iron uptake systems under various *in vitro* conditions (44). The experiment was initiated with *P. aeruginosa* PAO1 wild type populations and ran for 200 consecutive days. Although the QS

**TABLE 1** Mutations within each QS system

| System | Gene | Description | No. of clones |
|---|---|---|---|
| Las | lasI-rsaL-lasR | Las signal, repressor, and receptor | 29 |
| | lasR | Las receptor | 6 |
| Rhl | rhlR | Rhl receptor | 5 |
| PQS | pqsABCD | PQS signal operon | 3 |
| | pqsE | PQS signal operon | 4 |
| | pqsR | PQS receptor | 21 |

systems were not the focus of this work, whole-genome sequencing of evolved clones revealed an accumulation of nonsynonymous mutations in all three QS systems, corroborating the notion that mutations in these systems are commonly favored in this species. In the first step, we conducted an in-depth genomic analysis on the types, sizes, and location of mutations found in the Las, Rhl and PQS systems. Next, we screened all mutants for four QS-regulated traits to examine which type of mutation lead to a loss of function versus a modulated QS response. The four QS-regulated traits are (i) proteases, used to digest extracellular proteins; (ii) pyocyanin, a broad-spectrum toxin and redox active molecule; (iii) rhamnolipid biosurfactants, needed for group-level motility; and (iv) the ability to form surface-attached biofilms. Finally, we picked a subset of QS mutants with apparent QS-regulon modifications and investigated whether these mutations altered the gene expression of QS regulators and downstream-regulated traits.

## RESULTS

**Mutational patterns across the three QS regulons of *P. aeruginosa*.** The QS mutants originated from a previously described experimental evolution study (44). *P. aeruginosa* PAO1 was evolved in nine different casamino acids media with varying iron availabilities and environmental viscosities, for 200 consecutive days, in a total of 216 independent populations (24 populations per environmental condition). More details on the experimental setup are described in the Methods. While the initial study focused on the evolution of iron acquisition systems, whole-genome sequencing of 119 evolved clones (isolated at the end of the experiment) revealed that 61 clones had a total of 68 mutations in at least one of their QS systems (see Table 1 for an overview and Table S1 for individual clones). These QS mutants were isolated across all nine medium conditions (Table S2). Given that the 61 clones originated from 46 different populations, the mutational patterns we describe here mostly reflect independent evolutionary events. We detected 29 large-scale deletions (>4,903 bp), 30 single nucleotide polymorphisms (SNPs), and 9 microindels representing small deletions (max 12 bp) in the genes within the Las, Rhl and PQS systems. Most mutations were observed within the Las system ($n = 35$), followed by the PQS ($n = 28$) and Rhl ($n = 5$) systems.

**Mutations within the Las regulon.** Most mutations in the Las regulon entail large-scale deletions ($n = 29$, 82.9%, ranging from 4,903 to 65,969 bp) in which the Las signal synthase (*lasI*), the negative repressor (*rsaL*), and the Las receptor (*lasR*) were deleted, in addition to other genes (Fig. 1A). Our analysis could detect large deletions due to a very high sequencing coverage in this region of the genome (Table S3). In contrast to the large-scale deletions, we only found a small number of SNPs ($n = 6$, 17.1%) in the *lasR* receptor, of which five were in the same region of the DNA-binding domain (DBD) (Fig. 1B). The single mutant with an SNP at a different location also had a mutation in the PQS system.

**Mutations within the Rhl regulon.** In total, we found five SNPs in the gene coding for the Rhl receptor (*rhlR*). Two of the five mutants also have SNPs in the PQS system. Although the numbers are too few to obtain a conclusive pattern on the location of mutations, we found that the three clones which only had *rhlR* mutated all have SNPs in the ligand-binding site of *rhlR* (Fig. 1B). Meanwhile, the two *rhlR*-PQS double mutants have their SNPs outside the ligand-binding site.

**Mutations within the PQS regulon.** Out of the 26 clones with mutations in the PQS system, 19 have single mutations within the PQS regulon, 2 have double mutations within the PQS regulon, and 5 share one other mutation in either the Las or the Rhl regulon. In

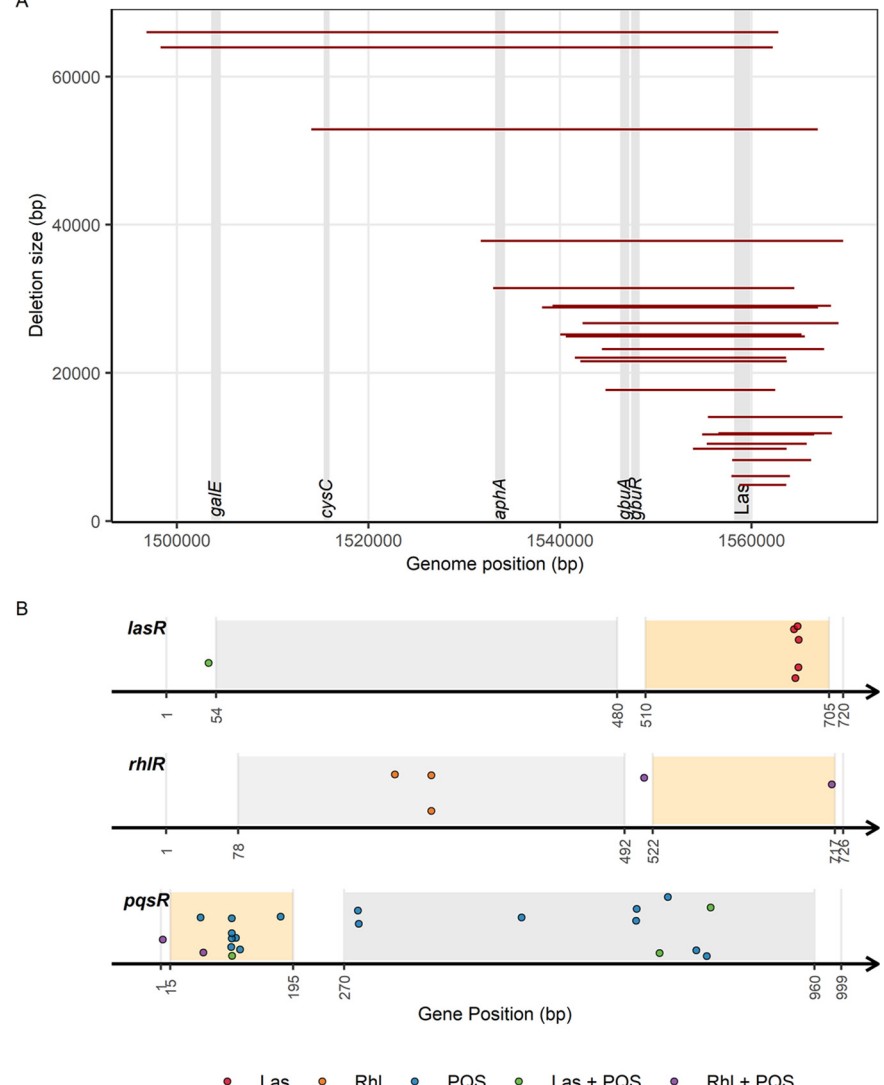

**FIG 1** Experimentally evolved mutations in the QS systems of *P. aeruginosa*. (A) Size and position of large-scale deletions that include *lasI*, *rsaL*, and *lasR* of the Las system. Shaded area represents position of genes in the genome. (B) Evolved mutations (single-nucleotide polymorphisms [SNPs] and microindels) in the genes encoding the receptors of Las (*lasR*, 720 bp), Rhl (*rhlR*, 726 bp) and PQS (*pqsR*, 999 bp) systems. Each dot represents the position of the mutated nucleotide within each gene. Gray and orange areas represent the ligand (i.e., signal)-binding and DNA-binding domains, respectively.

total, there were 28 mutations comprised of 2 SNPs and 1 microindel in the PQS signal operon (*pqsABCD*), 4 SNPs in the *pqsE* gene, and 13 SNPs and 8 microindels in the gene encoding the PQS receptor (*pqsR*). When mapping the mutations in *pqsR*, we found that the SNPs and microindels occurred both in the DNA- and ligand-binding domains (LBD) (Fig. 1B).

**QS system-specific mutations drive divergence in the production of QS-regulated traits.** Next, we explored how mutations in the Las, Rhl, and PQS systems are linked to growth and QS trait expression (proteases, rhamnolipids, pyocyanin, and biofilm). We grouped mutants into five categories: (i) clones with mutations in the Las receptor, *lasR*, and large-scale Las deletions (these two classes were combined because there was no difference in their phenotypes); (ii) clones with mutations in the Rhl regulon alone; (iii) clones with mutations in the PQS regulon alone; (iv) clones with mutations in the Las and the PQS regulons; and (v) clones with mutations in the Rhl and the PQS regulons. For the statistical analysis, we further included the ancestral wild type as a sixth category and determined whether there were significant differences in growth and QS-regulated trait production between the mutant categories and the wild type, as well as between the five mutant categories.

Our growth assay in LB medium revealed no significant difference in endpoint growth (i.e., yield) between any of the five mutant categories and the ancestral wild type (Fig. 2A, one-way analysis of variance [ANOVA], $F_{5,177} = 0.587$, $P = 0.710$). However, there were considerable differences in growth yield between evolved clones within certain mutant categories, especially among those with mutations in the Las regulon.

For proteases, we found significant differences in the production levels across the five mutant categories and the ancestral wild type (Fig. 2B, one-way ANOVA, $F_{5,156} = 20.388$, $P < 0.001$). All clones with mutations in the Las system (including the Las + PQS double mutants) had lower protease production compared to the ancestral wild type, with most clones having almost completely abolished production, similar to the lab-generated *lasR* mutant. Meanwhile, all clones with mutations in the Rhl system (including the Rhl + PQS double mutants) produced larger amounts of proteases than the ancestral wild type. This observation is diametrically opposite to the pattern seen in the lab-generated *rhlR* mutant, which does not produce proteases. Clones with mutations in the PQS system displayed a bimodal phenotypic profile: 14 produced almost no proteases, while 7 clones had a similar or higher protease production level compared to the ancestral wild type.

Pyocyanin production is significantly reduced in all mutant categories compared to the ancestral wild type (Fig. 2C, one-way ANOVA, $F_{5,191} = 70.212$, $P < 0.001$, all pairs tested with *post hoc* Tukey's honestly significant difference test [HSD] show $P_{adj} < 0.001$), but there are no significant differences between the mutant categories (all pairs tested with *post hoc* Tukey's HSD show $P_{adj} > 0.500$).

Rhamnolipid production was also significantly reduced in all mutant categories relative to the ancestral wild type (Fig. 2D, one-way ANOVA, $F_{5,191} = 49.003$, $P < 0.001$, all pairs tested with *post hoc* Tukey's HSD show $P_{adj} < 0.001$). But this time, we also observed significant differences in rhamnolipid production between the mutant categories (*post hoc* Tukey's HSD test, Las versus PQS and Las versus Las + PQS, $P_{adj} < 0.001$; Rhl versus Rhl + PQS, $P_{adj} = 0.009$). Clones with mutations in the PQS system stood out from the other categories because they showed enormous variability in rhamnolipid production, spanning the entire continuum from zero to levels almost identical to those of the ancestral wild type.

Finally, when looking at the ability of these clones to form surface-attached biofilms, we found significant differences in biofilm production between the five mutant categories and the ancestral wild type (Fig. 2E, one-way ANOVA, $F_{5,191} = 14.502$, $P < 0.001$). While clones with Las mutations showed significantly reduced biofilm formation compared to the ancestral wild type (*post hoc* Tukey's HSD test, Las versus wild type, $P_{adj} = 0.001$; Las + PQS versus wild type, $P_{adj} = 0.012$), clones with Rhl and PQS mutations were on average not different from the ancestral wild type (*post hoc* Tukey's HSD test, Rhl versus wild type, $P_{adj} = 0.859$; PQS versus wild type, $P_{adj} = 1.000$). However, we again observed enormous variability among PQS mutants: while some mutants showed extremely reduced biofilm formation, others invested considerably more into this trait compared to the ancestral wild type.

These findings suggest that mutations in the Las system spur broad-scale loss of function of QS traits, while mutations in the Rhl and PQS systems alter the QS-regulated trait expression patterns. To explore the apparent phenotypic segregation between mutant categories, we performed a principal-component analysis (PCA) incorporating all five phenotypes into a single analysis (Fig. 2F). We found that the evolved clones significantly clustered based on the mutant categories (permutational multivariate ANOVA [PERMANOVA]; $F_{4,60} = 28.167$, $P = 0.001$). When focusing on the loadings of the first two principal components (PCs) (i.e., vectors in Fig. 2F, Table S4), we identified two tradeoffs among the QS-regulated traits. PC1 yields a tradeoff between planktonic growth and biofilm formation as well as rhamnolipid production, meaning that evolved clones producing larger amounts of biofilm matrix components and rhamnolipids tend to grow less well in planktonic cultures. PC2 reveals a tradeoff between protease and pyocyanin production, indicating that evolved clones which produce higher levels of proteases produce lower levels of pyocyanin and vice versa. At the global level, we can conclude that modulation in the production of QS traits seems to be guided by tradeoffs, suggesting that maintaining or increasing the expression of one

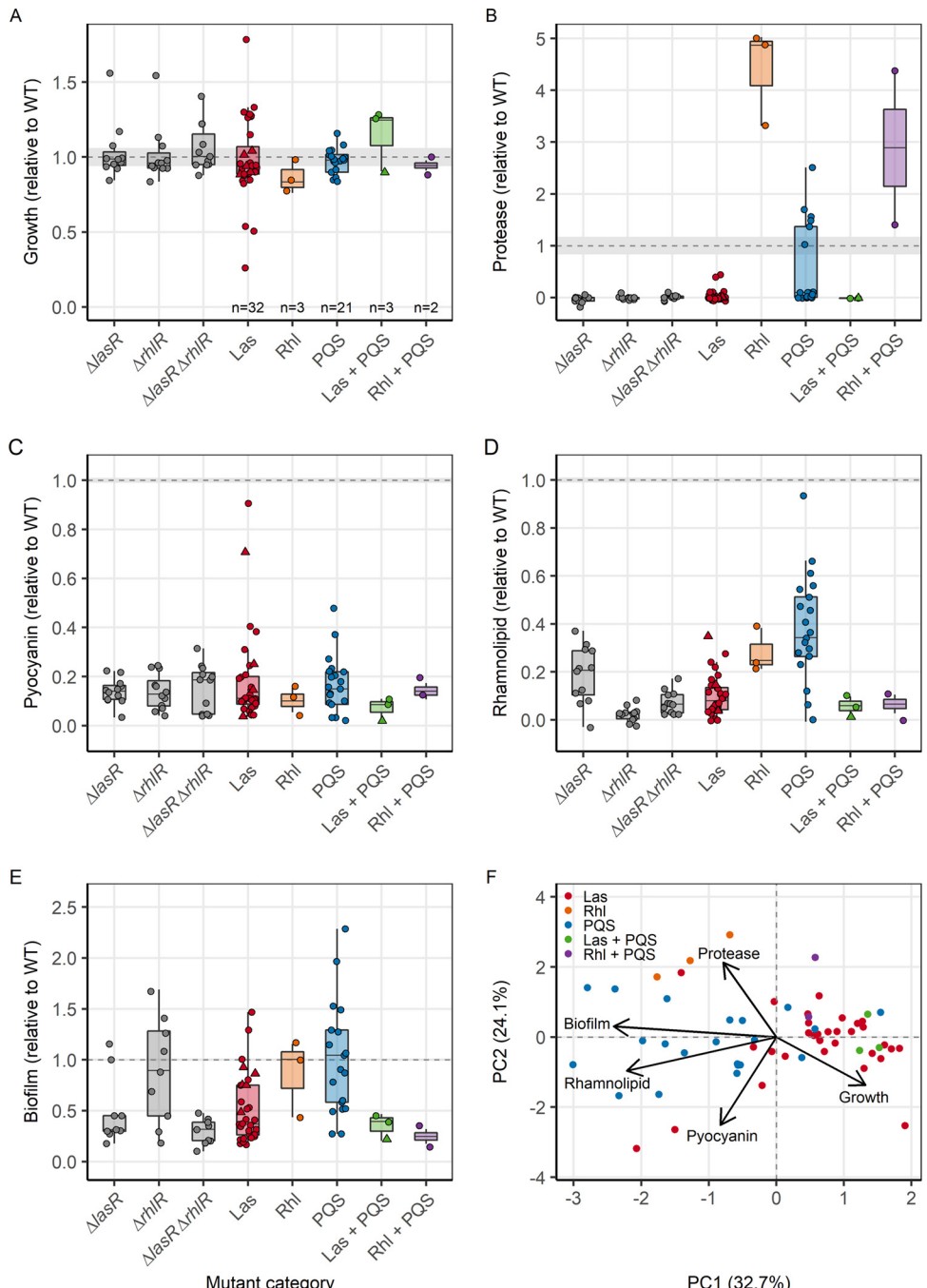

**FIG 2** Phenotypic profile of experimentally evolved *P. aeruginosa* isolates with mutations in the quorum-sensing (QS) regulon. Phenotypes of the 61 QS-mutants are expressed relative to the corresponding value of the ancestral PAO1 wild type strain (mean ± standard error, indicated as dashed lines and shaded areas, respectively). (A) Endpoint planktonic growth (optical density at 600 nm [$OD_{600}$]) in Lysogeny broth (LB) medium after 24 h. Production of four QS-regulated traits: (B) protease ($OD_{366}$), (C) pyocyanin ($OD_{691}$), (D) rhamnolipid ($\mu g/mL$) and (E) surface-attached biofilm ($OD_{570}/OD_{600}$). Data are shown for clones with mutations in either a single (Las, Rhl, or PQS systems) or multiple QS systems (Las + PQS, and Rhl + PQS). For Las-mutants, dots and triangles represent cases with large deletions and SNPs, respectively. Constructed QS mutants deficient in the production of either one of the two QS receptors, LasR ($\Delta lasR$), RhlR ($\Delta rhlR$), or both receptors ($\Delta lasR$-$\Delta rhlR$) were used as controls for production of QS-regulated traits in loss-of-function mutants. (F) Principal-component analysis (PCA) on growth and the production of the four QS-regulated traits. Each data point represents a single clone (with the average of at least three independent replicates).

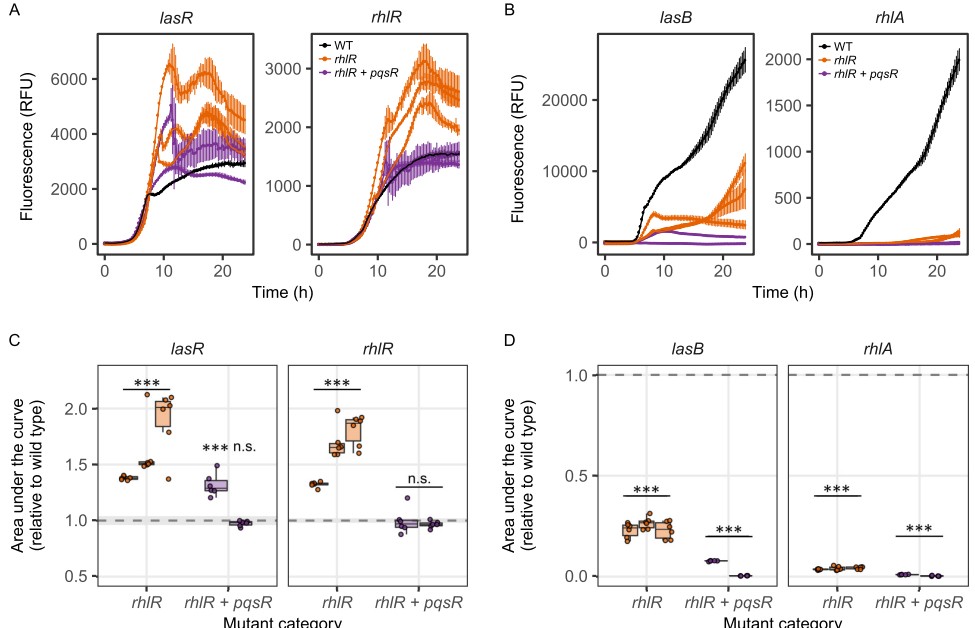

**FIG 3** Mutations in *rhlR* upregulates expression of the Las and Rhl receptors. Gene expression trajectories of (A) Las (*lasR*) and Rhl (*rhlR*) receptors, and (B) Las-regulated protease (*lasB*) and Rhl-regulated rhamnolipid (*rhlA*) in clones with mutations in *rhlR* (orange) and *rhlR* + *pqsR* (purple) shown as means ± standard deviation. Gene expression in PAO1 wild type strain was used as the reference control (black). Gene expression was measured as mCherry or green fluorescent protein (GFP) fluorescence and reported as fluorescence units, blank corrected by the background fluorescence of the wild type untagged strain. (C and D). Area under the gene expression trajectory curves in individual clones (represented by boxplots), relative to the gene expression in PAO1 wild type strain (dashed line at 1.0 ± standard deviation, depicted as shaded area). Data stem from 6 independent replicates per clone. Asterisks indicate whether area under the curve is significantly different from the PAO1 wild type strain (based on *post hoc* Tukey's honestly significant difference [HSD]; n.s., not significant; ***, $P < 0.001$).

QS trait is associated with a proportional reduction of another. Because QS modulations seem to be most marked among Rhl and PQS mutants, we focus more closely on these two QS systems in the next sections.

**Modulation of the Rhl regulon.** All three clones which had SNPs in the Rhl receptor, RhlR, had highly upregulated protease production and downregulated pyocyanin and rhamnolipid production, but retained wild type level formation of surface-attached biofilm. This phenotypic profile points toward QS regulon modulation, where the trait profile of these clones has changed. Here, we hypothesize that these phenotypic modulations should be reflected at the gene expression level. To test this, we used double fluorescent gene reporters to simultaneously measure the transcriptional gene expression activity of *rhlR* and the receptor of the upstream Las system, *lasR*, in these three clones over a growth period of 24 h in Lysogeny broth (LB) medium (Fig. 3A, Table S5). We found that mutations in *rhlR* significantly upregulated the expression of its own gene compared to the wild type strain (one-way ANOVA: $F_{2,33} = 54.950$, $P < 0.001$; *post hoc* Tukey's HSD test, $P_{adj} < 0.001$). However, this upregulation did not occur in the two clones which had *pqsR* mutations in addition to *rhlR* mutations (*post hoc* Tukey's HSD test, $P_{adj} = 0.971$). Curiously, we found that *lasR* expression was also significantly increased in clones with mutations in *rhlR* and in one of the two clones with mutations in both *rhlR* and *pqsR* (one-way ANOVA, $F_{2,33} = 22.554$, $P < 0.001$). The expression trajectory of *lasR* in the four overexpressing clones follows a cyclical pattern with two successive expression peaks at 10 and 18 h. The second peak coincides with the expression peak observed in *rhlR*. Taken together, our results reveal that point mutations in *rhlR* can lead to highly increased gene expression levels of the QS receptors RhlR and LasR.

Increased expression of QS receptors could lead to higher transcriptional regulator activity within the QS network and further translate to increased expression of the downstream QS genes. To test this hypothesis, we measured the expression of *lasB*, a protease regulated by both the Las and Rhl systems (45), and *rhlA*, part of the RhlAB

rhamnolipid operon regulated by the Rhl system (Fig. 3B) (46). We found no support for our hypothesis because *lasB* expression was reduced in all of the five clones (one-way ANOVA, $F_{2,33} = 1,204.4$, $P < 0.001$; *post hoc* Tukey's HSD test for all pairs, $P_{adj} < 0.001$). Similarly, we found strongly reduced expression of *rhlA* in all five clones, with some of the expression levels being close to zero (one-way ANOVA, $F_{2,33} = 8,046.8$, $P < 0.001$; *post hoc* Tukey's HSD test for all pairs, $P_{adj} < 0.001$). These findings show that increased expression of the LasR and RhlR QS receptors does not translate into increased expression of the two downstream regulated QS-traits. For *rhlA*, our gene expression results are compatible with the phenotypic data, as all mutants showed greatly reduced rhamnolipid production. For *lasB*, our gene expression results suggest that proteases other than LasB, such as LasA and AprA (47, 48), might be responsible for the observed high protease production at the phenotypic level.

**Modulation of the PQS regulon.** Our phenotypic screening, as shown in Fig. 2, revealed that mutations in the PQS system result in the most variable changes in the QS-regulated traits, with several clones showing an upregulation of these traits. Here, we focus on the 21 clones which have mutations only in the PQS system to explore whether evolved QS phenotypes depend on the mutated gene within the PQS locus, and whether there are tradeoffs, where the upregulation of one QS trait results in the downregulation of another one. Accordingly, we split the clones based on the mutated sites: PqsABCD (responsible for biosynthesis of the PQS precursor molecule), PqsE (alternative ligand that binds to RhlR), PqsR (PQS receptor), and those with mutations in more than one PQS site and re-ran our phenotypic analysis (Fig. 4A to E). We found that the sample size was too small for most categories to reliably establish relationships between phenotypes and mutational patterns. However, when conducting a PCA with all clones, we found that the evolved phenotypic profiles differed significantly between the mutation sites within the PQS regulon (PERMANOVA; $F_{3,20} = 2.712$, $P = 0.031$, Fig. 4F). We further observed two tradeoffs among traits (Fig. 4F, Table S6). First, clones with higher levels of protease production and biofilm formation produced less pyocyanin (Fig. S1A and B). Second, clones with higher levels of biofilm formation had lower growth in planktonic culture (Fig. S1C).

Next, we focused on the clones with mutations in *pqsR*, which represented the most frequent mutant type and showed the highest variability for most phenotypes. Especially, the bimodal profile of protease production shown in Fig. 2 is prevalent among the *pqsR* mutants (Fig. 4B). Here, we tested whether the divergent trajectories across *pqsR* mutants are linked to the location of the mutations (ligand- versus DNA-binding domain) or the type of mutations with regard to their deleterious effects (missense versus frameshift/deletions) (Table S7). However, we found that neither of these two factors can explain the bimodal protease production profiles (Fisher's exact test: mutation location, $P = 1$; mutation type, $P = 0.608$). Additionally, we tested whether mutations within the *pqsE* and *rhlR* genes affect residues that are implicated in the interaction between these two proteins, given that PqsE is an alternative ligand that binds to RhlR (25–29). We found no mutations that could be associated with such protein interactions.

## DISCUSSION

Because it is an important human pathogen, the evolution of *P. aeruginosa* has been studied in numerous contexts, with extensive genetic adaptation being repeatedly observed in diverging environments such as human cystic fibrosis (CF) lungs, animal infection models, as well as in natural habitats and *in vitro* experimental evolutions (6, 16, 18, 34, 38–41). The quorum-sensing system, a global three-unit regulatory network, is often among the most commonly mutated pathways. QS controls the expression of up to 10% of the genes in *P. aeruginosa*, many of which encode virulence factors and secreted public goods (11, 13, 14, 16, 49). It remains unclear why mutations in the QS systems are consistently favored across different environments. QS mutants could arise and spread due to (i) disuse of the regulon, (ii) cheating on the cooperative benefits of QS, or (iii) modulation of one or several of the three systems. Here, we used a set of 61 experimentally evolved QS mutants (with mutations in the three systems, Las, Rhl, and PQS) to examine these three scenarios. We found clear

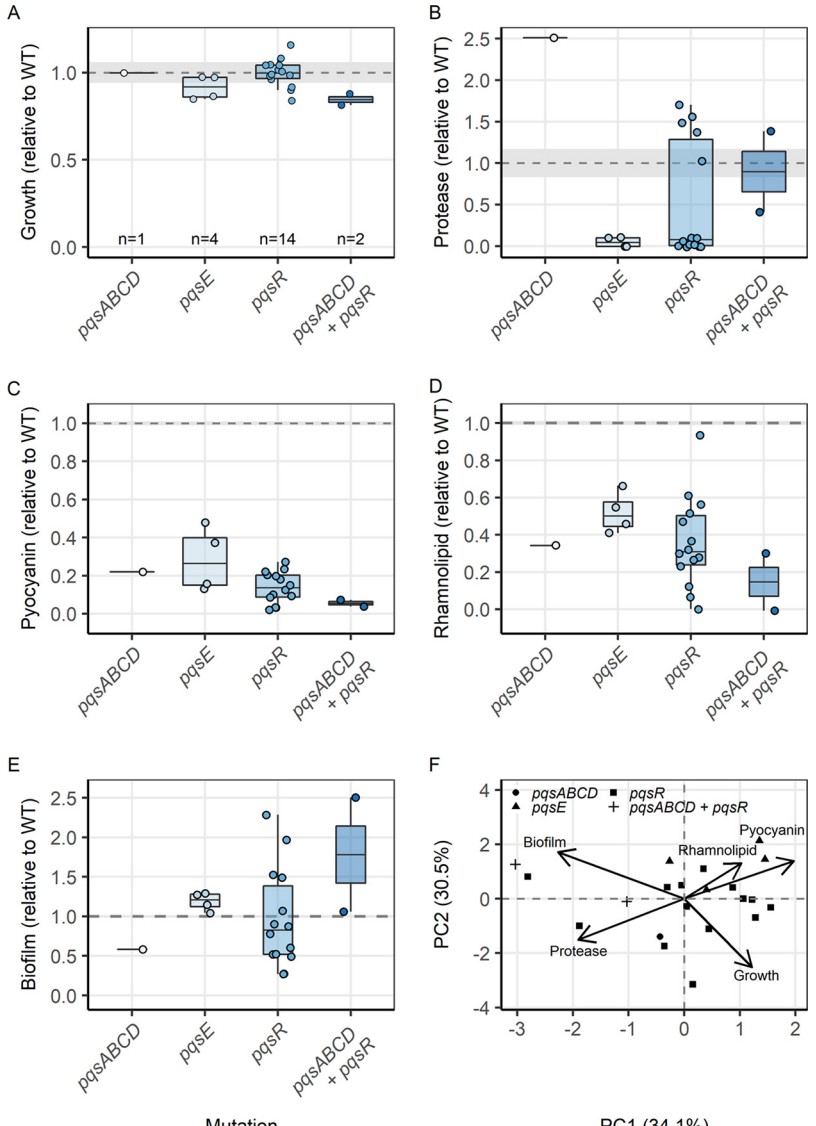

**FIG 4** Phenotypes in *Pseudomonas* Quinolone Signal (PQS) mutants and tradeoffs in the production of QS-regulated traits. Phenotypes of the 21 PQS-mutants are expressed relative to the corresponding value of the ancestral PAO1 wild type strain (mean $\pm$ standard error indicated as dashed lines and shaded areas, respectively). (A) Endpoint planktonic growth (optical density at 600 nm, $OD_{600}$) in LB medium after 24 h, and production of four QS-regulated traits: (B) protease ($OD_{366}$), (C) pyocyanin ($OD_{691}$), (D) rhamnolipid ($\mu$g/mL), and (E) surface-attached biofilm ($OD_{570}/OD_{600}$). (F) PCA on the production of growth and four QS-regulated traits reveals significant clustering of mutant types and significant tradeoffs (opposing vectors) between certain phenotypes. Each data point represents the average measure of at least three independent replicates per clone.

distinction between the QS systems in how mutations affected the production of QS-regulated traits. While mutations in the Las system generally resulted in the loss of QS function (supporting the first and second scenarios), mutations in the Rhl and PQS systems resulted in an altered trait production profile and regulatory network modulations (supporting the third scenario).

The mutational and phenotypic patterns observed in the Las mutants strongly point toward loss of QS function as opposed to regulon modulation. Our findings are partially in contrast with those of previous studies which showed that Las mutants can retain QS activity, partly through re-wiring the QS network (6, 32, 33). We suggest that the difference in our findings is driven by the fact that we predominantly found large-scale deletions of the Las system, where the signal synthase (*lasI*), repressor (*rsaL*), and regulator (*lasR*) are deleted, leading

to an overall loss of QS function. There is increasing evidence that large-scale deletions of the Las system are common in *in vitro* experimental evolution (40, 50, 51), but might have been overlooked in the past due to computational challenges of identifying them in draft genomes produced by short-read sequencing. Although we also used short-read sequencing, our coverage was high so that large deletions could be identified with high confidence (Table S3). Given that these large-scale deletions occurred many times independently, they must have an adaptive advantage, and they have previously been associated with both cheating and loss due to disuse in the process of medium adaptation (40, 50, 51).

In our case, all Las mutants (both large-scale deletions and point mutations) emerged from PAO1 wild type populations that had been experimentally evolved in Casamino acid medium, predominantly consisting of digested amino acids, an environment in which classic QS-regulated traits such as proteases and rhamnolipids are not necessarily needed. Thus, disuse rather than cheating is the more plausible explanation for the selective spread of Las mutants. Moreover, these mutants likely had an additional selective advantage because Las mutations were associated with increased production of the siderophore pyoverdine (Fig. S2), which was beneficial in the context of the initial study (44). In conclusion, whether the loss of the Las system is caused by disuse or cheating is often context-dependent and spurred by the relative costs and benefits of the QS system in the respective environment (31, 52, 53).

In contrast, we found evidence for QS regulon modulation in the Rhl mutants. These mutants arose at a much lower frequency than the Las mutants, similar to previous studies (54, 55). Both the phenotypic profiles and gene expression analyses revealed that there is no complete loss of QS function, but rather a change in the expression of the QS-regulated trait profile. At the phenotypic level, the regulon modulation is characterized by upregulating protease production and downregulating pyocyanin and rhamnolipid production, while retaining the ability to form surface-attached biofilms at levels similar to those of the ancestral wild type. This points toward possible decoupling of certain elements from the QS regulon. A straightforward explanation would be that mutations in *rhlR* abolish the production of traits directly controlled by the Rhl system, such as phenazines and rhamnolipids, while maintaining the traits which are predominantly under the control of the hierarchically superior Las system, such as proteases. However, our data contradict this explanation as we found largely increased protease production and significantly increased *rhlR* and *lasR* gene expression in the Rhl mutants (Fig. 3). One possible explanation is that mutations in *rhlR* give rise to RhlR receptor variants which, upon binding to the signal, show altered transcriptional affinities to specific promoter-binding sites. Indeed, we found asymmetrically altered transcriptional activities in which *rhlR* expression was increased while *rhlA* expression decreased. The high *rhlR* expression is intriguing, and we propose two independent mechanisms for this observation. First, RhlR can play a negative role in the expression of its own gene (56), and mutations in *rhlR* can potentially alleviate this inhibitory effect. Second, mutated RhlR receptors might be less effective in repressing PQS activity, and therefore, the increased *rhlR* expression could be a result of the positive feedback response by the PQS system.

Our analysis of the PQS mutants revealed that they are quite common (similar in frequency to the Las mutants) and show clear evidence for QS regulon modulation rather than loss of QS function. Unlike the Rhl mutants, which all had similar changes in their trait production profile, PQS mutants show a heterogeneous profile, with phenotypic tradeoffs between some of the QS-regulated traits (Fig. 2, Fig. S1). The clearest negative relationship among PQS mutants occurred between their ability to form surface-attached biofilms and their growth potential in liquid culture. This represents a classic tradeoff between a sessile versus a planktonic lifestyle. Weaker tradeoffs occurred between pyocyanin versus protease production and pyocyanin production versus biofilm formation. The latter tradeoff is perhaps hardest to explain because pyocyanin is an important compound involved in biofilm metabolism (57). However, it is important to note that the divergent phenotypes can occur within the same population (Fig. S1), which leads to two interesting possibilities for why natural selection could favor such diversification. First, the divergent phenotypes might occupy different ecological niches, to which they become well adapted. Second, mutants with diverging phenotypes may each specialize in the production of a set of QS-regulated traits

and share these traits with the other specialists at the group level. In conclusion, the modulation of QS network in the Rhl and PQS mutants is intricate and further genetic work is required to elucidate the exact regulatory trajectories which drive the altered trait expression profile among these mutants. Additionally, a deeper understanding on how mutations in upstream regulatory elements which control the QS network (e.g., VqsM, AlgR, and Vfr [3, 58, 59]) is crucial to elucidate the fine-tuning of the QS regulon modulation.

Finally, we also had a low number of clones with mutations in two QS systems (Las + PQS, $n = 3$; Rhl + PQS, $n = 2$). The phenotypes of the Las + PQS mutants seem to be dominated by mutations in the Las system, leading to loss of QS function. Similarly, the phenotypes of Rhl + PQS mutants point more toward loss of QS function rather than regulon modulation. For example, while single Rhl or PQS mutants produce rhamnolipids and can form surface-attached biofilms, albeit at varying levels, the double mutants show abolished phenotypes. The damping of QS modulation in double mutants is also observed at the gene expression level, where the Rhl + PQS mutants did not show increased *rhlR* expression as observed for the Rhl single mutants (Fig. 3). This indicates that an active PQS system is required for the regulon modulation to function in Rhl mutants. Our results are in line with those of previous studies reporting that mutation in the PQS regulon can result in a partial loss of Rhl activity, possibly through the disruption of an alternative signaling molecule, PqsE, which is recognized by RhlR (25, 26, 29, 33).

In conclusion, our results reinforce the view that QS is under selection not only in infections but also in *in vitro* experimental evolution. We show that mutational patterns and the resulting phenotypes are complex. While mutations in the Las system typically are associated with loss of QS function, we found that mutations in the Rhl and PQS systems lead to regulon modifications. Here, we support the hypothesis that QS systems can evolve and be rewired to match prevailing conditions in the laboratory and the host. Our study yields the first indications that modulation may drive strain diversification and adaptation to different ecological niches and may perhaps also foster mutualistic interactions between emerging strains (60). The next goal would be to understand how regulon modulation affects QS trait plasticity and bacterial fitness across different environments and what the consequences of QS modulation are for virulence in hosts.

## MATERIALS AND METHODS

**Bacterial strains.** We analyzed a collection of 61 experimentally evolved *P. aeruginosa* clones from Figueiredo et al. (44) (Table S1). All clones have a common ancestor, the PAO1 wild type strain (ATCC 15692). We grouped the evolved clones based on the mutations accumulated in either a single (Las, Rhl, or PQS systems) or multiple QS systems. For the growth and QS-phenotype screening assays, we additionally used the ancestral PAO1 wild type strain and three isogenic QS mutants constructed from the same PAO1 background. The isogenic QS mutants are deficient in the production of either one of two QS receptors, LasR ($\Delta lasR$) and RhlR ($\Delta rhlR$), or both receptors ($\Delta lasR$-$\Delta rhlR$). These are loss-of-function mutants and were used as controls for the screening of QS-regulated trait production.

To track gene expression in a subset of mutated clones ($n = 5$), we engineered double fluorescent transcriptional reporter fusions to measure the simultaneous expression of (i) *lasR-gfp* and *rhlR-mCherry*, and (ii) *lasB-gfp* and *rhlA-mCherry*. A single copy of the double reporter construct was chromosomally integrated in the experimentally evolved clones at the neutral attTn7 site using the mini-Tn7 system (61). Detailed step-by-step cloning protocol is described elsewhere (62). We used *Escherichia coli* CC118 $\lambda$pir for all intermediary steps in our cloning work (see Table S8 for a full list of non-experimentally evolved strains and plasmids used).

**Experimental evolution.** The protocol of the experimental evolution study is described in detail elsewhere (44). Briefly, experimental cultures were initiated with ancestral *P. aeruginosa* PAO1 and evolving populations were propagated for 200 consecutive days, during which approximately 1,200 generations occurred. Populations were cultured in 200 $\mu$L of casamino acid medium (5 g/L casamino acids, 1.18 g/L K$_2$HPO$_4$·3H$_2$O, 0.25 g/L MgSO$_4$.7H$_2$O, 25-mm HEPES buffer) in 96-well plates under shaking conditions (170 rpm) at 37°C. Two environmental parameters were manipulated independently: iron availability (no FeCl$_3$ added, 2 $\mu$m FeCl$_3$, or 20 $\mu$m FeCl$_3$ to achieve conditions of low, intermediate, and high iron availability, respectively) and environmental viscosities (0%, 0.1%, or 0.2% [wt/vol] agar to represent low, mid, or high spatial structure, respectively). Thus, there was a 3-by-3 full-factorial design of 9 environmental conditions in which *P. aeruginosa* populations evolved. There were 24 replicated populations per condition, resulting in a total of 216 independently evolving populations. Cultures were transferred to fresh media at a 1:10,000 dilution every 48 h (approximately $2 \times 10^5$ cells were transferred), while a 50 $\mu$L sample of each culture was also mixed with an equal volume of 85% glycerol and stored at −80°C. At the end of the evolution experiment, diluted cultures were spread onto LB agar plates supplemented with 20 $\mu$M FeCl$_3$ and 20 random clones were picked per population for phenotypic screening. Although the environmental conditions were important for the initial study design, they do not serve a specific purpose for the current study. It is important to note that QS mutants arose in all 9 environments (Table S2).

**Bioinformatical analysis.** In the previous study (44), the 119 evolved clones sequenced were drawn from nine growth conditions they evolved in. Because the initial study focused on the evolution of pyoverdine (siderophore) production, the clones were selected for sequencing in such a way as to represent the diversity in pyoverdine production phenotypes observed. Selected clones were sequenced on the Illumina NovaSeq6000 platform (paired-end, 150 base-pair reads). Among the sequenced clones, 61 (51.2%) had mutations in genes of the QS regulons (Table S1).

Detailed bioinformatics analyses are described elsewhere (44). In brief, the genomes of the ancestral strain and the evolved clones were assembled and aligned against the *P. aeruginosa* PAO1 reference genome. All the variants present in both ancestral and evolved clones were called and mutations which were already present in the ancestral strain used to initiate the evolution experiment were filtered out. Single-nucleotide polymorphisms and microindels (small insertions and deletions) were detected by aligning the obtained reads to the *P. aeruginosa* PAO1 reference genome using the Burrows-Wheeler Aligner "mem" algorithm followed by variant-calling with BCFTOOLS and annotation with SnpEFF. To detect large-scale deletions, the assembled genomes were analyzed by creating a "Mapping Graph Track" with CLC Genomic Workbench. Subsequently, every portion with an average coverage of less than $0.5\times$ was marked as a deletion (window size = 5 bp). Each deletion was then visually confirmed with Integrated Genomics Viewer.

For this study, we conducted an additional bioinformatic analysis. To map the position of SNPs and microindels within each QS gene, we compared the sequenced genome of the single evolved clones to the *P. aeruginosa* PAO1 reference genome on www.pseudomonas.com. Moreover, we used the published protein database of the QS signal-receptor complexes on InterProScan to obtain the classification of protein families and domains and extracted the information on the amino acid residues of the ligand- and DNA-binding domains of the Las, Rhl, and PQS transcriptional regulator complexes. Finally, to evaluate mutational hot spots, we mapped the positions of the evolved mutations to the reference gene sequences of *lasR*, *rhlR*, and *pqsR*.

**Growth measurements.** For all experiments, we pre-cultured single clones from freezer stocks in 6 mL lysogeny broth at 37°C, 220 rpm for 18 h. Prior to experiments, we washed overnight cultures twice with 0.8% NaCl and adjusted to an optical density at 600 nm ($OD_{600}$) of 1. To measure growth, we inoculated cells from overnight pre-cultures into 1.5 mL of fresh LB medium to a final starting $OD_{600}$ of 0.01 in 24-well plates and incubated them at 37°C for 24 h under shaken conditions (170 rpm). The purpose of this experiment was to obtain a proxy for fitness for all evolved clones relative to the ancestor in a standard medium, where the QS network is induced, but not essential (62). After 24 h, we measured growth as $OD_{600}$ in a microplate reader (Tecan Infinite M-200, Switzerland).

**Pyocyanin production.** To measure pyocyanin production, we collected the bacterial cultures after 24 h of growth in LB medium (described above) in 2 mL reaction tubes. We thoroughly vortexed and centrifuged them at $12,000 \times g$ for 10 min to pellet bacterial cells. We then transferred the cell-free supernatants to fresh 2 mL reaction tubes. For each clone, we transferred four 200 $\mu$L aliquots of the cell-free supernatant to 96-well plates, and quantified pyocyanin by measuring the optical density at 691 nm in a microplate reader. LB medium was used as a blank control.

**Rhamnolipid production via drop collapse assay.** We used the drop-collapse assay to measure rhamnolipid production. We collected the cell-free supernatants of bacterial cultures grown in LB medium as described above. For each clone, we plated 5 $\mu$L of the cell-free supernatant on the lids of 96-well plates and measured the droplet surface area after 1 min (63). Surface tension decreases with increasing concentrations of biosurfactant in the supernatant, resulting in the collapse of droplets (64). We took pictures of the lids and measured droplet surface area with the Image Analysis Software ImageJ. LB medium was used as a blank control. To quantify biosurfactant production based on droplet surface area, we made a calibration curve with a known range of synthetic rhamnolipid (Sigma-Aldrich, Buchs SG, Switzerland) concentrations (ranging from 0 to 0.2 g/L) and measured their respective droplet surface area.

**Protease production.** We used the azocasein assay to measure protease production. For this, we inoculated cells from overnight pre-cultures into 1.5 mL casein medium (5 g/L casein from bovine milk, 1.18 g/L $K_2HPO_4 \cdot 3H_2O$, 0.25 g/L $MgSO_4 \cdot 7H_2O$) to a final starting $OD_{600}$ of 0.01 in 24-well plates, and incubated the cultures at 37°C for 48 h under shaken conditions (170 rpm). After 48 h, we transferred the bacterial cultures to 2-mL reaction tubes, vortexed them thoroughly, and centrifuged at $12,000 \times g$ for 10 min to pellet bacterial cells. Next, we transferred the cell-free supernatants to fresh 2 mL reaction tubes. We first treated 40 $\mu$L aliquots of cell-free supernatants with 120 $\mu$L phosphate buffer (50 mM [pH $\approx$ 7.5]) and 40 $\mu$L azocasein (30 mg/mL), and subsequently incubated them at 37°C for 30 min. We stopped the reaction with 200 $\mu$L trichloroacetic acid (20%). We centrifuged treated supernatants at $12,000 \times g$ for 10 min and collected and transferred the fresh supernatants into new 96-well plates. We quantified protease production as optical density at 366 nm in a microplate reader. Casein medium treated with azocasein was used as a blank control. All medium components were purchased from Sigma-Aldrich (Buchs AG, Switzerland).

**Biofilm measurements.** We used a crystal violet assay to measure the ability of evolved clones to form surface-attached biofilms. We prepared overnight pre-cultures of single clones from freezer stocks in 200 $\mu$L LB medium in 96-well plates and incubated them at 37°C under static condition for 24 h. We measured the growth of pre-cultures at $OD_{600}$ using a microplate reader. Then, we diluted the pre-cultures to a starting $OD_{600}$ of 0.01 in 100 $\mu$L fresh LB medium in a 96-well round-bottom plate (no. 83.3925.500, Sarstedt, Germany) and incubated at 37°C under static conditions for 24 h. After this, we carefully transferred the cultures to a fresh flat-bottomed 96-well plate and measured growth at $OD_{600}$ in a microplate reader. We added 100 $\mu$L of 0.1% crystal violet to each well of the round-bottomed plate to stain the surface-attached biofilm and incubated the plates at room temperature for 30 min. Then, we carefully washed the wells twice with double-distilled water to remove the crystal violet solution and left them to dry at room temperature for 15 min. Next, we added 120 $\mu$L of dimethyl sulfoxide (DMSO) to each well to solubilize the stained biofilm and incubated the reaction at room temperature for 20 min. Finally, we measured optical density at 570 nm in a microplate reader, and

the production of surface-attached biofilm was quantified by calculating the "Biofilm Index" ($OD_{570}/OD_{600}$) for each well (65). LB medium treated with crystal violet and DMSO was used as a blank control.

**Gene expression measurement.** We inoculated fluorescent gene reporter cells from overnight cultures into fresh LB medium to a final starting $OD_{600}$ of 0.01 in individual wells on 96-well plates. Plates were incubated at 37°C in a microplate reader. We measured mCherry fluorescence (excitation = 582 nm, emission = 620 nm), green fluorescent protein (GFP) fluorescence (excitation = 488 nm, emission = 520 nm), and growth ($OD_{600}$) every 15 min (after a shaking event of 15 s) over a duration of 24 h. To remove background fluorescence, we measured the mean fluorescence intensity of the untagged PAO1 wild type strain in the mCherry and GFP channels over time and subtracted these values from the measured mCherry and GFP fluorescence values of the QS gene reporter strains at each time point.

**Statistical analysis.** We performed all statistical analyses with R studio (version 3.6.1). For all data sets, we consulted Q-Q plots and used the Shapiro-Wilk test to examine whether the residuals were normally distributed. We used one-way ANOVA and *post hoc* Tukey's HSD to compare growth and QS-regulated traits between the different mutant categories, and between the mutant categories and the ancestral wild type. We performed a principal component analysis (PCA) on the clonal phenotypes using the vegan package in R (version 2.5-7) (66). We further tested whether mutant categories differed in their evolved QS trait profiles using PERMANOVA. To compare gene expression trajectories, we fitted a parametric growth model (logistic model) in R and extracted the area under the curve (AUC) of each clone. Then, we used a one-way ANOVA to compare the AUC between the mutant categories.

**Data availability.** All raw data is available on Figshare: DOI:10.6084/m9.figshare.21191254. The sequencing data can be found in the European Nucleotide Database (https://www.ebi.ac.uk/ena/), under accession no. PRJEB45376.

## SUPPLEMENTAL MATERIAL

Supplemental material is available online only.

**FIG S1**, EPS file, 0.2 MB.
**FIG S2**, EPS file, 0.1 MB.
**TABLE S1**, DOCX file, 0.03 MB.
**TABLE S2**, DOCX file, 0.01 MB.
**TABLE S3**, DOCX file, 0.02 MB.
**TABLE S4**, DOCX file, 0.01 MB.
**TABLE S5**, DOCX file, 0.02 MB.
**TABLE S6**, DOCX file, 0.01 MB.
**TABLE S7**, DOCX file, 0.01 MB.
**TABLE S8**, DOCX file, 0.02 MB.

## ACKNOWLEDGMENTS

We thank Richard Allen for help with statistical analysis.

This work was funded by the European Research Council (ERC) under the European Union's Horizon 2020 Research and Innovation program (grant agreement no. 681295), the Swiss National Science Foundation (grant no. 31003A_182499), and a grant from the University Research Priority Program "Evolution in Action."

P.J., A.R.T.F., and R.K. designed the study. P.J. performed the experiments and analyzed the data. P.J., A.R.T.F., and R.K. interpreted the data and wrote the paper.

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
