## [Reviewer comments · mSystems]

Evolution of quorum sensing in *Pseudomonas aeruginosa* can occur via loss of function and regulon modulation

Priyanikha Jayakumar, Alexandre Figueiredo, and Rolf Kümmerli

Corresponding Author(s): Rolf Kümmerli, University of Zurich

Review Timeline:

Submission Date:	April 12, 2022
Editorial Decision:	June 6, 2022
Revision Received:	August 31, 2022
Accepted:	September 12, 2022

Editor: Alejandra Rodríguez-Verdugo

Reviewer(s): The reviewers have opted to remain anonymous.

Transaction Report:

DOI: <https://doi.org/10.1128/msystems.00354-22>

June 6, 2022

Prof. Rolf Kümmerli
University of Zurich
Department of Quantitative Biomedicine
Zurich 8057
Switzerland

Re: mSystems00354-22 (Evolution of the quorum sensing systems in *Pseudomonas aeruginosa* can involve both loss of regulon function and network modulation)

Dear Prof. Rolf Kümmerli:

Thank you for submitting your manuscript to mSystems. We have completed our review and I am pleased to inform you that, in principle, we expect to accept it for publication in mSystems. However, acceptance will not be final until you have adequately addressed the reviewer comments.

The two reviewers and I agree this is a relevant study with interesting findings. Both reviewers have made excellent suggestions for improvement that should be addressed before we can accept your work for publication. One of the main points that was brought up is that it would be helpful to provide more information about the original experiment from which the mutants originated (i.e. Figueiredo, Wagner and Kümmerli, 2021). Another important point is that the figures need to be improved (increase the font, add labels...). Finally, the reviewers raised important questions/thoughts that should be addressed in the discussion. I look forward to receiving a revised manuscript.

Preparing Revision Guidelines

Sincerely,

Alejandra Rodríguez-Verdugo

Editor, mSystems

Journals Department
Reviewer comments:

Reviewer #1 (Comments for the Author):

This is an interesting manuscript by Jayakumar and colleagues about the changes in quorum sensing network regulation of *Pseudomonas aeruginosa* that occur in an experimental evolution system. The authors describe the accumulated mutations and show different consequences of the mutants on the three QS circuits of *P. aeruginosa*.

Issues to consider:

1. There is inadequate exposition of the various elements of the PQS system -- particularly the differences in pqsABCD vs pqsR vs pqsE and how these different components feed into the QS hierarchy. In the same vein, there's been a significant amount of recent work describing putative interactions between PqsE and RhIR (primarily from the Paczowski and Bassler labs) and should be included in the discussion of how the mutations might modulate QS.

1b. Do the mutations in rhIR or pqsE map to residues that have been implicated in the interaction between the proteins these genes encode?

2. The experimental evolution from which these isolates were obtained asked a question unrelated to QS. This is not a problem but does distinguish the present work from others who have performed experiments with *P. aeruginosa* to test how QS evolves (which generally have used a QS-dependent growth condition). Some discussion of how the experimental design might affect the observations would be helpful.

3. The finding of increased transcription of rhIR in the RhIR mutant backgrounds is really interesting. What's the possible mechanism? Is RhIR known to regulate the transcription of its own gene?

4. A significant issue was that the text in several figures was very difficult, almost impossible, to read!

Reviewer #2 (Comments for the Author):

Overall, Jayakumar et al focuses on the evolution of quorum sensing systems in *P. aeruginosa*, and how evolution can involve both loss of regular function and network modulation. The premise is well set-up with clear background and hypotheses, and many of the findings are interesting. Some of the experimental detail is lacking, and the origins of the 61 best-followed clones is not clear, although there is more information cited in recent publications where evolved isolates that are the focus of this manuscript are taken from. It is interesting that the QS mutants are common, especially large Las deletions, and that there are patterns linking the mutations in different QS systems and phenotypes. The trade-off between protease and pyocyanin production and biofilm formation or growth is surprising and interesting. The following constructive comments are intended to improve this interesting study.

1. Line 25: QS systems are described as a common mutational target - please comment on how much is known about how this is affected by environment? (e.g. both in the lab and in situ in environmental or infection conditions? How about polymicrobial, static or better mixed conditions?)

2. Line 65-66: loss-of-function mutations that reduce virulence are often found in infectious contexts, and maybe allow for a chronic infection to continue without attracting immune attention... furthermore, molecules described as virulence factors serve functions that are not related to pathogenicity, such as the electron shuttling capacity of pyocyanin. Please keep this in context.

3. At times the lack of experimental detail while referencing Figueriedo et al 2021 made it hard to follow the set-up and the story. A few key items: Line ~150 - were the cultures shaking or static? How often were the cultures transferred, and how much of a bottleneck was introduced at each step? (e.g. 1:10 passages, or 1:100?). At what timepoint were the 119 evolved clones taken, at the end of the 200 days?

4. On Line 156, nine environments are mentioned... is tis triplicates of the three iron conditions? Are the 119 sequenced clones

arising across those 9 environments somewhat equally? Can the 61 isolates being followed in this study be treated independently, if some arose in the same replicate culture?

5. Some frequently used terms in the manuscript are poorly defined or understood, including virulence (without a host, hard to define, from the bacterial perspective may serve important purposes outside pathogenicity, like pyocyanin as an alternative electron shuttle), QS network, and social phenotypes, and social portfolio in P.a. Even the title contains some of these phrases and may benefit from simplification or clarification.

6. Fig 2, Fig 3, Fig 4 and Supp Fig 1: please add units to the axes. It is especially surprising to see that increased pyocyanin production, which should help as an electron shuttle in a low oxygen environment, does not go together with increase in biofilm formation capacity.

7. How did the approach in this study enable detection of large deletions where other papers with short reads were not able to? Were data assembled for the clones before detection of large deletions? Consider including coverage maps

8. Conclusions - This section contains some long and complex sentences with many references, consider simplifying and adding some bigger picture thoughts.

Minor:

a. Line 431, typo, context > contexts

b. Line 510: analysis on > of

c. Line 554: hard to follow what "this" refers to

d. How was 24 hour growth endpoint chosen? Are kinetics at earlier timepoints distinct??

Reviewer comments:

Reviewer #1:

This is an interesting manuscript by Jayakumar and colleagues about the changes in quorum sensing network regulation of *Pseudomonas aeruginosa* that occur in an experimental evolution system. The authors describe the accumulated mutations and show different consequences of the mutants on the three QS circuits of *P. aeruginosa*.

1. Thank you for the summary and for your interest in our work. Below we addressed your comments together with a detailed explanation of the implemented changes.

Issues to consider:

1. There is inadequate exposition of the various elements of the PQS system -- particularly the differences in pqsABCD vs pqsR vs pqsE and how these different components feed into the QS hierarchy. In the same vein, there's been a significant amount of recent work describing putative interactions between PqsE and RhIR (primarily from the Paczowski and Bassler labs) and should be included in the discussion of how the mutations might modulate QS.

- 1.1 There are two separate concerns raised here. Below we address them one by one.

- 1.1.1 We agree with the reviewer that our explanation on the different components of the PQS system and how they feed into the QS hierarchy was incomplete. We have revised Lines 65-80 to include more details on the PQS system and how it is wired with the upstream, Las and RhI systems. We have also described the different components of PQS system on Lines 274-277.

- 1.1.2 Thank you for emphasizing the interactions between PqsE and RhIR. There has been a lot of work in the recent years to validate the mechanism of how PQS activates RhI system – either through PqsE directly, or PqsE-induced ligand. We have included this aspect at several points in our manuscript, precisely on Lines 78-80 (introduction), Lines 291-294 (results) and Lines 385-388 (discussion).

1b. Do the mutations in rhIR or pqsE map to residues that have been implicated in the interaction between the proteins these genes encode?

- 1.1.3 Thank you for your interest regarding this point. We mapped the positions of the mutations in *rhIR* and *pqsE* but none of them are within the residues implicated in the interaction between RhIR and PqsE. We have now mentioned this aspect on Lines 291-294.

2. The experimental evolution from which these isolates were obtained asked a question unrelated to QS. This is not a problem but does distinguish the present work from others who have performed experiments with *P. aeruginosa* to test how QS evolves (which generally have used a QS-dependent growth condition). Some discussion of how the experimental design might affect the observations would be helpful.

1.2 Thank you for the comment. The original experimental design focused on the evolution of iron-acquisition strategies, particularly via the siderophore pyoverdine, under various *in vitro* conditions. Although pyoverdine production was thought to be largely independent from QS, we observed that mutation in the QS network is associated with increased pyoverdine production in most of the environments. We have included the pyoverdine phenotypes of the evolved clones in the new Fig. S2, and discussed the relationship between the study design and QS evolution on Lines 329-333.

3. The finding of increased transcription of *rhIR* in the *RhIR* mutant backgrounds is really interesting. What's the possible mechanism? Is *RhIR* known to regulate the transcription of its own gene?

1.3 Thank you for raising this point. We have now proposed two independent mechanisms that could explain the increased *rhIR* activity in the *RhIR* mutant background on Lines 352-356.

4. A significant issue was that the text in several figures was very difficult, almost impossible, to read!

1.4 We apologize for the small label size in the initial figures. We have now enlarged and re-sized the text in all our revised figures.

Reviewer #2:

Overall, Jayakumar et al focuses on the evolution of quorum sensing systems in *P. aeruginosa*, and how evolution can involve both loss of regular function and network modulation. The premise is well set-up with clear background and hypotheses, and many of the findings are interesting. Some of the experimental detail is lacking, and the origins of the 61 best-followed clones is not clear, although there is more information cited in recent publications where evolved isolates that are the focus of this manuscript are taken from. It is interesting that the QS mutants are common, especially large *Las* deletions, and that there are patterns linking the mutations in different QS systems and phenotypes. The trade-off between protease and pyocyanin production and biofilm formation or growth is surprising and interesting. The following constructive comments are intended to improve this interesting study.

2 Thank you for the summary and your constructive feedback on our work. Below we individually addressed each of your comments together with a description of the implemented changes.

1. Line 25: QS systems are described as a common mutational target - please comment on how much is known about how this is affected by environment? (e.g. both in the lab and in situ in environmental or infection conditions? How about polymicrobial, static or better mixed conditions?)

2.1 Thank you for the comment. We have now elaborated and addressed some of the environmental differences where QS mutants were isolated from on Lines 81-90.

2. Line 65-66: loss-of-function mutations that reduce virulence are often found in infectious contexts, and maybe allow for a chronic infection to continue without attracting immune attention... furthermore, molecules described as virulence factors serve functions that are not related to pathogenicity, such as the electron shuttling capacity of pyocyanin. Please keep this in context.

2.2 Thank you for the comment. We have addressed both concerns raised and revised our text on Lines 60-63 and Lines 69-74 to specify the potential adaptive benefits of QS loss and the role of QS-regulated traits outside the context of infections.

3. At times the lack of experimental detail while referencing Figueriedo et al 2021 made it hard to follow the set-up and the story. A few key items: Line ~150 - were the cultures shaking or static? How often were the cultures transferred, and how much of a bottleneck was introduced at each step? (e.g. 1:10 passages, or 1:100?). At what timepoint were the 119 evolved clones taken, at the end of the 200 days?

2.3 We agree with the reviewer that some key aspects of the original study were lacking. We have addressed all the questions raised here and added more details on the experimental setup on Lines 127-136 (Results) and Lines 421-439 (Materials and Methods). The reader now obtains a full overview on the methods of the original study.

4. On Line 156, nine environments are mentioned... is tis triplicates of the three iron conditions? Are the 119 sequenced clones arising across those 9 environments somewhat equally? Can the 61 isolates being followed in this study be treated independently, if some arose in the same replicate culture?

2.4 There are several concerns raised here which we addressed individually as follow:

2.4.1 We thank the reviewer for bringing to our attention that the nine environments were not clearly explained. Three iron availabilities and three environmental viscosities were combined in a full-factorial design, with nine different independent growth conditions. Each condition had 24 independently evolving populations. We have clarified this on Lines 127-131 and Lines 426-432.

2.4.2 A total of 119 clones were sequenced and they originated equally from the 9 growth conditions that they evolved in (Figueiredo, Wagner and Kümmerli, 2021). Out of the 119 clones, 61 had mutations in the QS regulon and they arose in all 9 growth conditions. We have described this on Lines 128-138 and tabulated the number of clones isolated from each evolved condition in Table S2.

2.4.3 Most of the clones can be treated as independent replicates, as the 61 clones originated from 46 different population. Moreover, we found that clones isolated from the same population often differed in their phenotypes (e.g., PQS mutants in Fig. S1), indicating that these clones also followed independent evolutionary trajectories. We have now mentioned this on Lines 136-138.

5. Some frequently used terms in the manuscript are poorly defined or understood, including virulence (without a host, hard to define, from the bacterial perspective may serve important purposes outside pathogenicity, like pyocyanin as an alternative electron shuttle), QS network, and social phenotypes, and social portfolio in *P.a.* Even the title contains some of these phrases and may benefit from simplification or clarification.

2.5 Thanks for pointing this out. In the revised version of our manuscript, we have either excluded unclear terms or explained them more explicitly. (i) We have removed the term social phenotypes. (ii) We have changed 'social portfolio' to 'QS-regulated trait profile'. (iii) We have simplified the title. (iv) We have better explained the QS network in *P. aeruginosa*. (v) We have explained the various roles of pyocyanin. (vi) We agree that we cannot define virulence in our *in vitro* study. But we have now explained what virulence factors are in the context of infections and mentioned their additional roles outside infections.

6. Fig 2, Fig 3, Fig 4 and Supp Fig 1: please add units to the axes. It is especially surprising to see that increased pyocyanin production, which should help as an electron shuttle in a low oxygen environment, does not go together with increase in biofilm formation capacity.

2.6 Thank you for the comment. There are two separate concerns raised here. Below we address them individually.

2.6.1 Thank you for bringing our attention to the unclear axis labels. The phenotypic assays in Fig. 2, Fig. 4, and Fig. S1 were presented as values relative to the wild type. We have now clarified this on the axis labels and included the units for all phenotypes measured in the respective figure legends. We have also included axis label for gene expression fluorescence in Figure 3.

2.6.2 We agree with the reviewer that it is surprising to see a trade-off between pyocyanin production and biofilm formation, given the importance of pyocyanin in low-oxygen environments. We now discuss this aspect on Lines 364-371. We propose that diversification can happen within a population with strain becoming specialized in different traits. Thus, a high pyocyanin producer could become embedded into a biofilm produced by the high biofilm matrix producers.

7. How did the approach in this study enable detection of large deletions where other papers with short reads were not able to? Were data assembled for the clones before detection of large deletions? Consider including coverage maps

2.7 As the reviewer pointed out, it is not always easy to detect large structural variations from short-read sequencing. For this reason, many microbial evolution-and-resequencing studies often only focus on SNPs and small indel calling. In contrast, we could detect large deletions in our analysis due to a very high sequencing coverage. We have explained this in detail on Lines 322-323. Additionally, we present Table S3 that shows an overview of the detection confidence using a gene (*aphA*) that was deleted in 8 out of the 29 clones bearing large-

scale deletions. This table shows that we had very high confidence in detecting large deletions.

8. Conclusions - This section contains some long and complex sentences with many references, consider simplifying and adding some bigger picture thoughts.

2.8 Thank you for the feedback and suggestions. We agree with the reviewer and felt that our entire discussion would benefit from clarifications. This is the reason why we have revised the entire discussion text to improve clarity and conciseness.

Minor:

a. Line 431, typo, context > contexts

We have implemented the change.

b. Line 510: analysis on > of

We have implemented the change.

c. Line 554: hard to follow what "this" refers to

We have revised our text in this section following comment 2.8 above.

d. How was 24 hour growth endpoint chosen? Are kinetics at earlier timepoints distinct??

We do not have kinetics data for comparison. Cultures are at stationary phase after 24 hours, which means that we compared yield across clones (now specified on Lines 178-179). The 24h timepoint was chosen because we used the same growth cultures in LB to measure yield, as well as pyocyanin and rhamnolipid.

September 12, 2022

Prof. Rolf Kümmerli
University of Zurich
Department of Quantitative Biomedicine
Zurich 8057
Switzerland

Re: mSystems00354-22R1 (Evolution of quorum sensing in *Pseudomonas aeruginosa* can occur via loss of function and regulon modulation)

Dear Prof. Rolf Kümmerli:

Thank you for carefully addressing each of the comments from the two reviewers. The manuscript has been strengthened by these changes and hopefully will have a wider impact in the field. The manuscript is now ready for publication! Thank you for the privilege of reviewing your work.

Your manuscript has been accepted, and I am forwarding it to the ASM Journals Department for publication. For your reference, ASM Journals' address is given below. Before it can be scheduled for publication, your manuscript will be checked by the mSystems production staff to make sure that all elements meet the technical requirements for publication. They will contact you if anything needs to be revised before copyediting and production can begin. Otherwise, you will be notified when your proofs are ready to be viewed.

Publication Fees:

If you would like to submit a potential Featured Image, please email a file and a short legend to msystems@asmusa.org. Please note that we can only consider images that (i) the authors created or own and (ii) have not been previously published. By submitting, you agree that the image can be used under the same terms as the published article. File requirements: square dimensions (4" x 4"), 300 dpi resolution, RGB colorspace, TIF file format.

We recognize that the video files can become quite large, and so to avoid quality loss ASM suggests sending the video file via <https://www.wetransfer.com/>. When you have a final version of the video and the still ready to share, please send it to mSystems staff at msystems@asmusa.org.

Sincerely,

Alejandra Rodríguez-Verdugo
Editor, mSystems

Journals Department
Fig. S2: Accept
Table S3: Accept
Table S7: Accept
Table S8: Accept
Table S2: Accept
Table S1: Accept
Table S6: Accept
Table S5: Accept
Table S4: Accept
Fig. S1: Accept